# OptTTA: Learnable Test-Time Augmentation for Source-Free Medical Image Segmentation Under Domain Shift

**Devavrat Tomar**[1]                                                    DEVAVRAT.TOMAR@EPFL.CH
[1] *Signal Processing Laboratory 5 (LTS5), EPFL, Switzerland*

**Guillaume Vray**[1]                                                    GUILLAUME.VRAY@EPFL.CH

**Jean-Philippe Thiran**[1,2,3,4]                              JEAN-PHILIPPE.THIRAN@EPFL.CH
[2] *University of Lausanne (UNIL), Switzerland*
[3] *Radiology Department, Centre Hospitalier Universitaire Vaudois (CHUV), Switzerland*

**Behzad Bozorgtabar**[1,3,4]                                  BEHZAD.BOZORGTABAR@EPFL.CH
[4] *Center for Biomedical Imaging (CIBM), Switzerland*

**Editors:** Under Review for MIDL 2022

## Abstract

As distribution shifts are inescapable in realistic clinical scenarios due to inconsistencies in imaging protocols, scanner vendors, and across different centers, well-trained deep models incur a domain generalization problem in unseen environments. Despite a myriad of model generalization techniques to circumvent this issue, their broad applicability is impeded as (i) source training data may not be accessible after deployment due to privacy regulations, (ii) the availability of adequate test domain samples is often impractical, and (iii) such model generalization methods are not well-calibrated, often making unreliable overconfident predictions. This paper proposes a novel learnable test-time augmentation, namely OptTTA, tailored specifically to alleviate large domain shifts for the source-free medical image segmentation task. OptTTA enables efficiently generating augmented views of test input, resembling the style of private source images and bridging a domain gap between training and test data. Our proposed method explores optimal learnable test-time augmentation sub-policies that provide lower predictive entropy and match the feature statistics stored in the BatchNorm layers of the pretrained source model without requiring access to training source samples. Thorough evaluation and ablation studies on challenging multi-center and multi-vendor MRI datasets of three anatomies have demonstrated the performance superiority of OptTTA over prior-arts test-time augmentation and model adaptation methods. Additionally, the generalization capabilities and effectiveness of OptTTA are evaluated in terms of aleatoric uncertainty and model calibration analyses. Our PyTorch code implementation is publicly available at https://github.com/devavratTomar/OptTTA.

**Keywords:** Learnable test-time augmentation, domain shift, medical image segmentation

## 1. Introduction

The common assumption of most deep models used for medical image segmentation is that training and test data distributions are alike. Nonetheless, this assumption can be easily broken in real-world situations, and deep models might encounter performance degradation when ported on a test environment that differs considerably from those used at training time due to variations in imaging protocols, scanner vendors, etc. Thus, many recent methods focus on improving model robustness trained on training data (a.k.a. source

domain) to generalize better in the new test environment (a.k.a. target domain). Several techniques, including unsupervised domain adaptation (UDA) methods (Tomar et al., 2021b; Vu et al., 2019; Chen et al., 2019b; Zhang et al., 2021; Bozorgtabar et al., 2019; Tomar et al., 2021a), and domain generalization (DG) approaches (Li et al., 2020; Dou et al., 2019) have been proposed; each formulates this problem differently. Nevertheless, there are still substantial practical barriers to using these techniques in clinical practice. Prior UDA and DG approaches require concurrent access to source and target samples or multiple source domains, often infeasible after model deployment due to privacy regulations arising from source data or when target data is scarce. Thus, a learning framework wherein only a source model is required to adapt itself to a new target domain without the source data is paramount for medical image segmentation. Recent methods have been proposed to tackle this issue based on source-free domain adaptation (Liu et al., 2021; Bateson et al., 2020) or test-time model adaptation (TTMA) (Sun et al., 2020; Nado et al., 2020). These methods often utilize self-training schemes with entropy minimization (Wang et al., 2021; Lee et al., 2013), test-time batch normalization (Nado et al., 2020), or additional auxiliary training networks (He et al., 2020; Karani et al., 2021; Valvano et al., 2021). Despite their practical success on minor domain shifts, those self-training techniques often produce incorrect predictions in the presence of large domain shifts leading to error accumulation during model adaptation as reported in previous works (Prabhu et al., 2021; Chen et al., 2019a; Jiang et al., 2020). Recently, test-time augmentation (TTA) methods (Wang et al., 2018; Isensee et al., 2018; Moshkov et al., 2020; Amiri et al., 2020; Wang et al., 2019) have shown promise in improving robustness and accuracy without retraining the model by aggregating predictions over multiple augmented versions of each test image. More recently, inspired by training-time policy search approaches (Cubuk et al., 2019; Lim et al., 2019; Hendrycks et al., 2020), test-time policy search methods (Lyzhov et al., 2020; Kim et al., 2020; Shanmugam et al., 2021) have been proposed for classification tasks to find static policies using either a greedy search algorithm (Lyzhov et al., 2020), an auxiliary module (Kim et al., 2020) to predict sample-specific loss, or a learnable aggregation strategy (Shanmugam et al., 2021). Nonetheless, they require policy search using a separate validation set, and learned augmentation policies might not be optimal for each test sample.

**Contributions.** To the best of our knowledge, (i) we propose the first learnable TTA policy, namely OptTTA, on the task of medical image segmentation tailored for alleviating large domain shifts. (ii) Despite existing TTA methods based on static policies, OptTTA dynamically selects optimal TTA policies producing transformed versions of test input, resembling the style of private source training images. (iii) OptTTA can be implemented in a streaming fashion via fine-tuning sub-policies sequentially for image volumes. (iv) Experiments on challenging multi-center and multi-vendor MRI datasets of various anatomies show OptTTA superiority against prior-arts. Further, we provide analyses for the TTA-based aleatoric uncertainty and model calibration to support the effectiveness of OptTTA.

## 2. Methods

This section describes our proposed method, OptTTA, for learning TTA policy on medical image segmentation under large domain shift using only a trained model on source data without requiring access to neither training source data nor all target data at once during

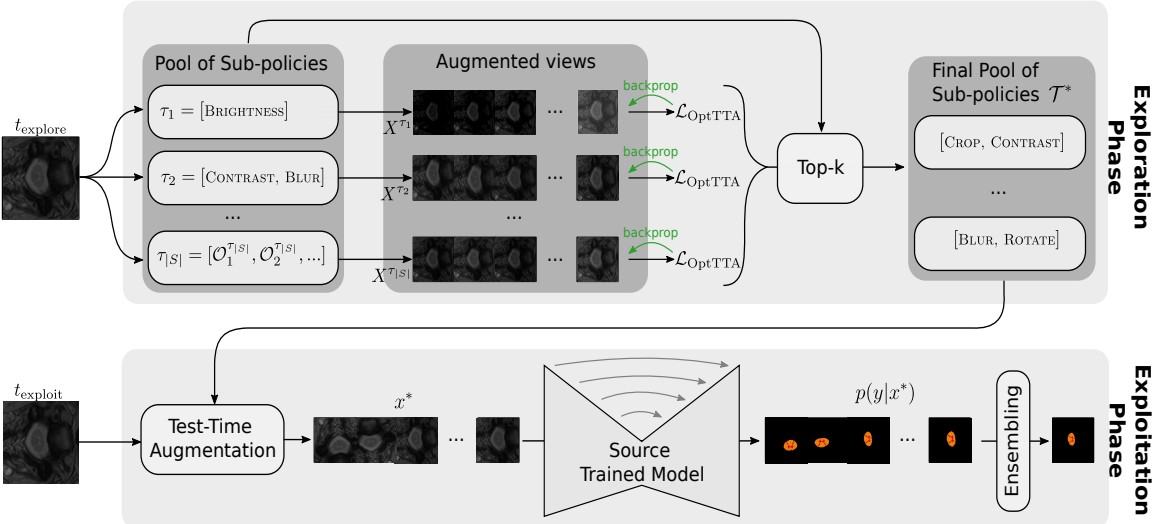

Figure 1: OptTTA involves two phases - (1) **Exploration**– All sub-policies in $S$ are optimized using gradient descent followed by the selection of top-k sub-policies as $\mathcal{T}^*$; (2) **Exploitation**– The sub-policies of $\mathcal{T}^*$ are fine-tuned in streaming fashion for the rest of the target image volumes $t_{\text{exploit}}$, followed by ensembling the predictions of the source model over multiple transformations of the test image volume.

inference. As shown in Fig. 1, OptTTA involves two phases: (1) **Exploration** and (2) **Exploitation**. In the Exploration phase, we search for data augmentation policies that perform well based on the evaluation criterion mentioned in Sec. 2.2.1 using a set of target image volumes $\mathbf{t}_{\text{explore}}$ without any segmentation labels. Once we find the optimal data augmentation policies in the Exploration phase, we fine-tune the same data augmentation policies for the rest of the target image volumes $\mathbf{t}_{\text{exploit}}$, one image volume at a time to generate multiple augmented views. The predictions of the source trained model on these optimal augmented views are then ensembled, yielding the final prediction. Here, we first introduce the policy search space (Sec. 2.1) comprising data augmentation operations followed by the TTA sub-policy evaluation criterion–$\mathcal{L}_{\text{OptTTA}}$ without ground-truth segmentation (Sec. 2.2.1). Finally, we describe a gradient-descent-based search algorithm for optimal TTA sub-policies in Sec. 2.2.2.

## 2.1. Policy Search Space

Let $\mathbb{O}$ be a set of image transformation operations $\mathcal{O} : \mathcal{X} \to \mathcal{X}$ on the image space $\mathcal{X}$. In particular, the list of transformations includes IDENTITY (I), GAMMA CORRECTION (G), GAUSSIAN BLUR (GB), CONTRAST (C), BRIGHTNESS (B), RESIZE CROP (RC), HORIZONTAL FLIP (HF), VERTICAL FLIP (VF), ROTATE (R). We parameterize each transformation $\mathcal{O}$ with its magnitude $\lambda$, sampled from a probability distribution $q_\theta$ with parameter $\theta$. Some transformations in $\mathcal{O}$ (i.e. HORIZONTAL FLIP, VERTICAL FLIP, ROTATE) do not have any learnable parameters. Let $S$ be a set of

sub-policies, where a sub-policy $\tau \in S$ consists of $N_\tau$ consecutive transformation operations from $\mathbb{O} : \{\mathcal{O}_n^\tau(x; \lambda_n^\tau) : n = 1, ..., N_\tau\}$, where each operation is applied sequentially as:

$$x_n = \mathcal{O}_n^\tau(x_{n-1}; \lambda_n^\tau) \tag{1}$$

where $x_0 = x$, $x_{N_\tau} = \tau(x)$ and $\lambda_n^\tau \sim q_{\theta_n^\tau}$. An example of a sub-policy is [Resize Crop, brightness, Horizontal Flip]. The final policy $\mathcal{T}$ is a collection of $N_{\mathcal{T}}$ sub-policies.

## 2.2. Evaluating and Optimizing TTA Sub-Policies

### 2.2.1. Evaluation Criterion

The main essence of our method relies on the observation that a source trained model outputs high confidence predictions (low entropy) and high accuracy for source-like images that also match the feature statistics stored in the BatchNorm layers of the pretrained model. Let $X^\tau$ denote the set of 2D augmented views of the target image volume $t$ generated using a sub-policy $\tau \in S$ by sampling the magnitude $\lambda_n^\tau$ of its operations $\{\mathcal{O}_n^\tau\}$ from a probability distribution $\{q_{\theta_n^\tau}\}$ with parameters $\{\theta_n^\tau\}$ using Eq. 1. We then define a test-time smoothing loss function over the outputs of the segmentation model on $X^\tau$ as follows:

$$\mathcal{L}(X^\tau) = \frac{1}{|X^\tau|} \sum_{x \in X^\tau} \mathcal{L}_{\text{ent}}(x) + \alpha_1 \mathcal{L}_{\text{bn}}(X^\tau) - \alpha_2 \mathcal{L}_{\text{cm}}(X^\tau) \tag{2}$$

where $\alpha_1$ and $\alpha_2$ are hyper-parameters, and the individual loss terms are described below.
**BatchNorm Statistics Loss ($\mathcal{L}_{\textbf{bn}}$).** This loss term acts as the feature distribution regularizer to penalize the distance between the statistics of network activations on the batch of augmented images $X^\tau$ and that of the private source data stored in the widely-used BatchNorm (BN) layers of the pretrained network.

$$\mathcal{L}_{\text{bn}}(X^\tau) = \sum_l \left( \left\| \mu_l\left(X^\tau\right) - \bar{\mu}_l \right\|_2^2 + \left\| \sigma_l^2\left(X^\tau\right) - \bar{\sigma}_l^2 \right\|_2^2 \right) \tag{3}$$

where $\mu_l\left(X^\tau\right)$ and $\sigma_l^2\left(X^\tau\right)$ are the batch-wise feature means and variances at the $l$-th BN layer for an input batch of augmented images $X^\tau$, and $\bar{\mu}_l$ and $\bar{\sigma}_l^2$ are the corresponding mean and variance parameters stored in the $l$-th BN layer.
**Conditional Entropy Loss ($\mathcal{L}_{\textbf{ent}}$).** This loss term is defined over the pixel predictions of the segmentation model on the input image $x$ and encourages high confidence predictions.

$$\mathcal{L}_{\text{ent}}(x) = -\sum_y p(y|x) \log p(y|x) \tag{4}$$

where $p(y|x)$ is the softmax output of the segmentation model on the input image $x$, and $y$ denotes model prediction spans over the segmentation classes.
**Entropy of Class Marginals ($\mathcal{L}_{cm}$)** Maximizing this loss term encourages the model predictions $\hat{p}(y) = \frac{1}{|X^\tau|} \sum_{x \in X^\tau} p(y|x)$ to be uniformly distributed over the segmentation classes as minimizing Eq. 4 alone may result in predictions converging to a single segmentation class. $\mathcal{L}_{cm}$ does not require any prior information about the segmentation class distribution.

$$\mathcal{L}_{\text{cm}}(X^\tau) = -\sum_y \hat{p}(y) \log \hat{p}(y) \tag{5}$$

### 2.2.2. OPTIMIZATION ALGORITHM

A sub-policy $\tau$ is evaluated by taking the expectation of Eq. 2 with respect to the random magnitudes of its augmentations. We then learn the distribution parameters $\theta^\tau = \{\theta_n^\tau : n = 1, ..., N_\tau\}$ associated with a sub-policy $\tau$ that minimize this expected loss.

$$\mathcal{L}_{\text{OptTTA}}^\tau(\theta^\tau, t) = \mathbb{E}_{X^\tau \sim \tau(t)}[\mathcal{L}(X^\tau)] \tag{6}$$

For estimating the gradients of $\mathcal{L}_{\text{OptTTA}}^\tau$ with respect to its corresponding probability distribution parameters $\theta^\tau$, we perform the **re-parametrization trick** by sampling magnitude $\lambda^\tau$ from a Uniform distribution as follows:

$$\lambda^\tau \sim \mu^\tau + \sigma^\tau \cdot \mathcal{U}(-\mathbf{1}, \mathbf{1}) \tag{7}$$

where $\theta^\tau = \{\mu^\tau, \sigma^\tau\}$, $\mathcal{U}(-\mathbf{1}, \mathbf{1})$ is $N_\tau$ dimensional Uniform distribution, and $\{\mu^\tau, \sigma^\tau\} \in \mathbb{R}^{N_\tau}$. Thus, $X^\tau$ becomes a function of $(\mu^\tau, \sigma^\tau)$ and the gradients of Eq. 6 are estimated as follows:

$$\nabla_{\theta^\tau} \widehat{\mathcal{L}_{\text{OptTTA}}^\tau} = \begin{bmatrix} \nabla_\mu^\tau \mathcal{L}(X(\mu^\tau, \sigma^\tau)) \\ \nabla_\sigma^\tau \mathcal{L}(X(\mu^\tau, \sigma^\tau)) \end{bmatrix} \tag{8}$$

We use the AdamW (Loshchilov and Hutter, 2018) gradient descent approach to optimize the parameters $\theta^\tau$ of the sub-policy $\tau$, summarized in the **Algorithm** (Appendix A).

### 2.3. Top-k Sub-Policies Selection and Test-Time Aggregation

During *Exploration*, we optimize every sub-policy in $S$ using the Algorithm described in Appendix A (Mode := explore) over target image volumes $t_{\text{explore}}$ and obtain the corresponding set of optimized sub-policies $S^*$. We observe that some of the optimized sub-policies in $S^*$ perform poorly with a large loss $\mathcal{L}_{\text{OptTTA}}$. Thus, we dynamically keep top $k$ sub-policies from $S^*$ having the $k$ lowest loss values in the final policy set $\mathcal{T}^*$ using the evaluation loss in Eq. 2 (*cf.* Table 5, Appendix D.2). In the *Exploitation* phase, we only fine-tune the optimal sub-policies in $\mathcal{T}^*$ using Algorithm in Appendix A (Mode := exploit) and generate augmented views of target image volumes $t_{\text{exploit}}$ one at a time in a sequential manner. For every sub-policy $\tau_i^* \in \mathcal{T}^* : i = 1, ..., k$, we generate $M$ augmented views of the target image volume $t$ and aggregate the predictions of the source trained model on these views:

$$\bar{p}(t) = \frac{1}{k \cdot M} \sum_{i=1}^{k} \sum_{j=1}^{M} p(y_{ij}|x_j^{\tau_i^*}) \tag{9}$$

where $x_j^{\tau_i^*} \in X^{\tau_i^*}$, which is sampled $M$ times independently from sub-policy $\tau_i^*$.

## 3. Experiments and Results

### 3.1. Datasets and Implementation Details

We measure the performance of OptTTA on three public multi-center, multi-vendor datasets. **Spinal Cord Grey Matter Segmentation (SCGM) dataset** (Prados et al., 2017). This dataset is collected from four different medical centers (1, 2, 3, 4) using four different MRI scanners annotated with two segmentation classes - Grey Matter, Spinal Cord Area.

**Heart Image Segmentation Dataset (M&Ms)** (Campello et al., 2021). This dataset contains 375 studies from six centers and four scanner vendors coded as A, B, C, and D with three segmentation classes - Left Ventricle, Right Ventricle, and Myocardium.

**Prostate MRI Segmentation Dataset** (Liu et al., 2020). This dataset is acquired from six different sites (A, B, C, D, E, F) with various imaging scanners annotated with the prostate area. Following the protocol of (Liu et al., 2020), we discard site C as it contains data from unhealthy patients. See Appendix B for more details about three MRI datasets.

**Implementation Details:** We adopt 2D U-Net architecture (Ronneberger et al., 2015) instead of the 3D version due to large variance in (volume shape, voxel spacing, and a number of axial slices from different centers) for the segmentation backbone trained on the source domain images using a combination of Dice and weighted cross-entropy losses. The source segmentation network is trained using data augmentation from set $\mathcal{O}$ (*cf.* Section 2.1), RMSprop optimizer with a learning rate of $10^{-5}$ (decay factor of 0.1 with 2 epochs patience), weight decay of $10^{-4}$, and momentum of 0.9 for $250K$ iterations. We set $\alpha_1 = 0.01$ and $\alpha_2 = 0.005$, respectively (*cf.* Table 4, Appendix D.2). We also set $|t_{\text{explore}}| = 1$, $k = 3$, $|N_\tau| = 5$, $M = 128$ for the main experiments of Table 1 (*cf.* Figs. 6, 7, and Table 6, Appendix D.2), and learning rate of $10^{-3}$, $\beta = (0.9, 0.999)$, weight decay of $10^{-4}$ for OptTTA Algorithm (Appendix A). All baselines are implemented in PyTorch (Paszke et al., 2019) and trained on NVIDIA GeForce RTX 3080 GPU. We use Hausdorff Distance (Dubuisson and Jain, 1994) (*cf.* Table 8, Appendix D.5.1) and Dice (%) as the evaluation metrics.

### 3.2. Comparison to State-of-the-Arts

Table 1 shows the quantitative comparison results (Dice (%)) with state-of-the-art methods: (a) UDA method including ADVENT (Vu et al., 2019) and ProDA (Zhang et al., 2021); (b) TTMA approaches including TENT (Wang et al., 2021), test-time normalization (BN) (Nado et al., 2020), where BN layers are updated with test domain statistics, and our new baseline (PL) that generates pseudo-labels by tuning a confidence threshold to optimize the model; and (c) TTA methods including greedy policy search (GPS*) (Lyzhov et al., 2020)[1], RandAug (Cubuk et al., 2020), and Vanilla test-time Augmentation (VA) (random crop, rotation, and flipping). Overall, OptTTA achieves the most significant average Dice improvement (9.2%, 22.5%, and 1.7% on Spinal Cord, Heart, and Prostate MRI datasets) compared to trained Source Model without adaptation. The TTMA baselines alleviate the reliance on the source domain and adapt to new test image volumes in an online fashion, but they often make incorrect predictions under substantial domain shifts leading to error accumulation and performance deterioration (Heart dataset). Similar observations hold for UDA methods that may encounter the deterioration of feature discriminability despite concurrent access to source and target samples. TTA methods marginally improve performance due to their static policies and limited search space. As shown in Fig. 2, OptTTA overcomes the above shortcomings by learning suitable augmentation policies and magnitudes of transformations necessary to alleviate domain shift and generate source-like augmented images, thus improving generalization capability on the test set. More qualitative are provided in Appendix D.5.

**Aleatoric Uncertainty and Model Calibration Analysis.** We analyze TTA-based aleatoric uncertainty with the lens of model calibration visualized with a *reliability* diagram

---

1. GPS is adapted for the segmentation task using $\mathcal{L}_{\text{OptTTA}}$ criterion in Sec. 2.2.1.

Table 1: Dice (%) results of mean(±std) on three datasets. The largest domain gap w.r.t. source domain is highlighted in red, and Bold values denote the best performances.

| | | Lower Bound | UDA | | TTMA | | | TTA | | | |
|---|---|---|---|---|---|---|---|---|---|---|---|
| Target site | # Volumes | Source Model | ADVENT | ProDA | BN | TENT | PL | VA | RandAug | GPS* | **OptTTA** |
| (Source Site 1) | | | | | | | **Spinal Cord** | | | | |
| 2 | 10 | 77.4±6.6 | 83.0±3.6 | **86.0±2.2** | 85.2±2.1 | 85.7±1.8 | 85.3±2.1 | 79.1±4.6 | 82.7±3.2 | 81.7±5.0 | 85.0±2.5 |
| 3 | 10 | 64.8±11.7 | 80.9±3.7 | 79.7±3.7 | 70.6±3.6 | 68.7±2.8 | 71.0±3.6 | 66.0±12.9 | 66.9±12.2 | 78.4±5.5 | **82.0±2.7**$^{\dagger}$ |
| 4 | 10 | 85.9±3.8 | 87.4±2.8 | **89.0±1.5** | 88.9±1.7 | 88.9±1.7 | 88.9±1.7 | 86.0±2.4 | 86.9±2.1 | 87.1±2.9 | 88.8±1.7 |
| Average | | 76.0±11.8 | 83.8±4.3 | 84.9±4.7 | 81.6±8.3 | 81.1±9.1 | 81.7±8.6 | 77.0±11.5 | 78.8±11.7 | 82.5±5.9 | **85.2±3.6** |
| (Source Sites A,B) | | | | | | | **Prostate** | | | | |
| D | 13 | 75.8±8.9 | 75.2±9.4 | 83.3±4.8 | 75.9±9.4 | 78.8±6.2 | 76.1±9.4 | 81.6±6.3 | 80.1±7.6 | 77.3±7.7 | **86.6±4.0**$^{\dagger}$ |
| E | 12 | 65.9±18.5 | 63.4±13.4 | **82.8±6.0** | 74.4±7.4 | 77.9±6.9 | 74.8±7.5 | 68.1±20.6 | 66.8±20.7 | 64.1±27.0 | 79.8±8.1 |
| F | 12 | 38.4±32.3 | 47.6±31.3 | 63.3±28.7 | 65.7±22.4 | 67.0±28.4 | 66.2±22.4 | 53.3±33.1 | 56.6±31.5 | 57.8±17.2 | **82.1±8.3** |
| Average | | 60.5±27.0 | 62.4±23.2 | 76.7±19.3 | 72.1±15.2 | 74.7±17.9 | 72.4±15.2 | 68.1±25.4 | 68.3±24.0 | 66.7±20.5 | **83.0±7.5**$^{\dagger}$ |
| (Source Site A) | | | | | | | **Heart** | | | | |
| B | 250 | 87.6±4.2 | 87.2±4.7 | 88.3±3.5 | 85.2±6.0 | 82.1±7.8 | 85.3±6.0 | 87.7±3.5 | 87.7±3.5 | 85.9±4.6 | **88.7±3.6**$^{\ddagger}$ |
| C | 100 | 85.5±4.4 | 83.9±5.8 | 86.4±3.5 | 82.9±6.3 | 79.9±7.7 | 83.0±6.3 | 87.2±3.6 | 87.1±3.7 | 85.6±6.1 | **87.8±3.4**$^{\ddagger}$ |
| D | 100 | 86.0±4.0 | 84.7±4.3 | 87.4±3.4 | 83.3±6.6 | 80.2±7.8 | 83.4±6.5 | 88.0±3.9 | 88.2±3.3 | 85.5±5.9 | **88.3±3.9** |
| Average | | 86.7±4.5 | 85.8±5.2 | 87.5±3.7 | 84.1±6.6 | 80.9±8.2 | 84.2±6.6 | 87.6±3.8 | 87.6±3.6 | 85.5±5.3 | **88.4±3.6**$^{\ddagger}$ |

[‡] $p < 0.005$, [†] $0.005 < p < 0.05$: A paired t-test with respect to the top results.

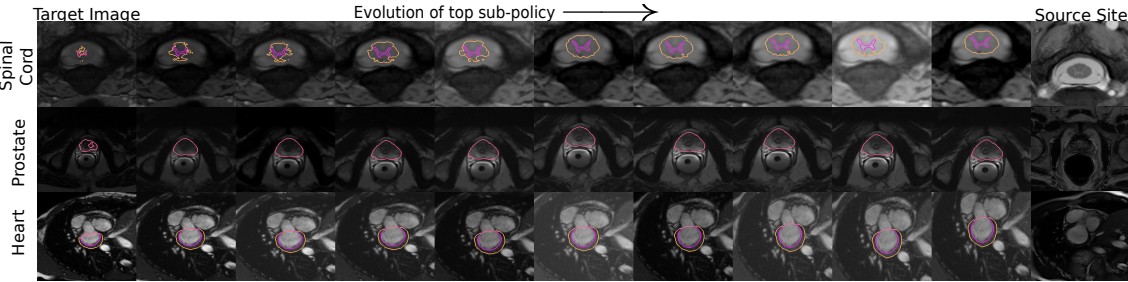

Figure 2: From left to right, starting from the initial augmented test image, we show the evolution of the top sub-policy on sample test images per dataset by a fixed step size of 110 iterations. The last column shows the corresponding source images.

(Niculescu-Mizil and Caruana, 2005). As shown in Fig. 3, different baselines' model performances are plotted against the binned confidence scores. Overall, Fig. 3 shows several compared baselines fail to output reliable confidence estimates matching the true underlying model performance when tested on sites other than the source site. Even when the model is inaccurate, these baselines make high confidence predictions making them unreliable. In contrast, OptTTA shows significantly better calibration for the segmentation classes. Our observations are supported with model uncertainty metrics, Brier score, and the Negative Log-Likelihood (NLL) (Gomariz et al., 2021) (*cf.* Appendix C) presented in Fig. 3 (b). OptTTA has a significantly lower Brier ($p < 0.005$) and NLL scores ($0.005 < p < 0.05$) than the second best, which correlates with the greater Dice score. As shown in Fig. 4, OptTTA outputs higher values of confidence map (i.e., lower aleatoric uncertainty) near the

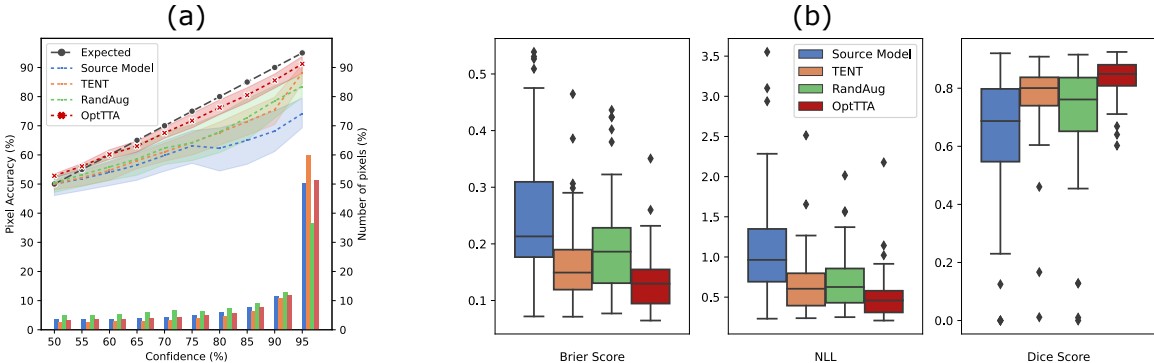

Figure 3: (a) Reliability diagrams for pixel-wise predictions and (b) Uncertainty metrics–
Brier and NLL metrics with Dice scores on the Prostate dataset.

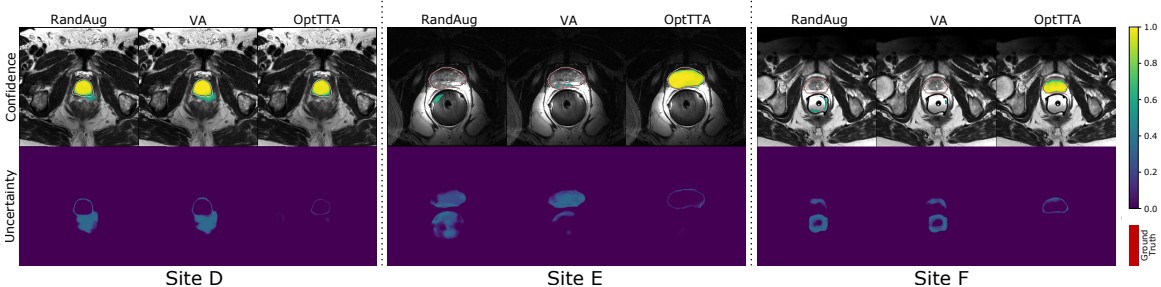

Figure 4: Comparison of the segmentation confidence and uncertainty of OptTTA against
other TTA baselines for the target sites D, E, F on the Prostate dataset.

boundary of the segmented prostate compared to other TTA baselines for the model trained
on source sites A, B and tested on target sites D, E, and F (*cf.* Appendix C.1).

## 4. Conclusion and Future Work

We propose a novel learnable TTA, OptTTA, for medical image segmentation tailored
for substantial domain shifts as opposed to the previous TTA methods that use static
augmentation policies. OptTTA offers a privacy-preserving solution, eliminating the need
for training data or extra model retraining by generating test-time augmented images in
the source style, enhancing segmentation performances by dynamically selecting optimal
policies compared to other baselines. Our method surpasses prior-arts by a large margin
and provides more reliable predictions.

OptTTA can be further extended to perform self-training based on the pseudo-labels
generated by our optimized TTA. Together with the release of our implementation, we
believe this work will inspire further research on model generalization under a significant
domain shift in clinical practice.

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

## Appendix A. Algorithm for OptTTA Optimization

Here we present the algorithm for optimizing a TTA sub-policy. The number of gradient steps for sub-policy $\tau$ is determined by the phase it is being updated. During Exploration, the gradient steps are kept much greater in Exploration phase ($N_{\text{grad}}^{\text{explore}} \sim 1000$) than in Exploitation phase ($N_{\text{grad}}^{\text{exploit}} \sim 100$).

**Input:**
Mode $\in$ {explore, exploit}
Trained segmentation model $p(y|x)$ on source data;
Target Image Volume $t_{\text{Mode}}$;
Sub-policy $\tau(\theta^\tau) : \{\mathcal{O}_n^\tau(x; \lambda_n^\tau) : n = 1, ..., N_\tau\}$;
Gradient Descent Steps $N_{\text{grad}}^{\text{Mode}}$; Learning rate $\eta$;
Batch size B of 2D augmented images for one iteration;
**Output:**
Optimized sub-policy $\tau^*$
**Initialization:**
  **for** $i \in \{1, ..., N_\tau\}$ **do**
  | $\theta_i^\tau \leftarrow \{0, 0.01\}$                              /* initialize with small number */
  **end**
**Optimization:**
  **for** $j \leftarrow 1$ *to* $N_{grad}^{Mode}$ **do**
  | $X \leftarrow \{\}$
  |   **for** $b \leftarrow 1$ *to* $B$ **do**
  |   | $a \leftarrow \mathcal{U}(t_{Mode})$                            /* sample 2D slice from $t_{\text{Mode}}$ */
  |   |   **for** $i \leftarrow 1$ *to* $N_\tau$ **do**
  |   |   | $\lambda_i^\tau \leftarrow \mu_i^\tau + \sigma_i^\tau \cdot \mathcal{U}(-1, 1)$        /* re-parametrization trick */
  |   |   | $a \leftarrow \mathcal{O}_i^\tau(a; \lambda_i^\tau)$               /* apply augmentations from $\tau$ */
  |   |   **end**
  |   | $X \leftarrow X \cup \{a\}$
  |   **end**
  | $\theta^\tau \leftarrow \theta^\tau - \eta \nabla_{\theta^\tau} \mathcal{L}(X)$                      /* defined in Eq. 8 */
  **end**
  **return** $\tau(\theta^\tau)$

### A.1. Analysis and Discussion of Computational Complexity

We conduct our experiments using NVIDIA GeForce RTX 3080 (6 optimization steps per second). Given a set of $\mathcal{S}$ sub-policies, the Exploration phase takes approximately $T_{explore} = 166.67 * |\mathcal{S}|$, in which a single sub-policy optimization (1000 iterations) takes around 166.67 seconds. Then, once we obtain the top-$k$ sub-policies from the Exploration phase, the prediction for each volume takes approximately $T_{exploit} = k * 16.67 + M * D * 0.007$ seconds, in which $M$ denotes the number of generated augmented views, and $D$ denotes the depth of the corresponding volume (number of 2D slices). The duration of the Exploitation phase on one sub-policy (100 iterations) takes 16.67 seconds, and the prediction cost of a

Table 2: The volume-wise computational time (seconds/volume) of OptTTA against several TTA baselines using NVIDIA GeForce RTX 3080. We report $T_{exploit}$ for OptTTA, $T_{explore}$ being negligible as $N$ tends to be large in practice. The times below are computed on the Spinal Cord (Prados et al., 2017) target site 3, Prostate MRI dataset (Liu et al., 2020) target site F and Heart dataset (Campello et al., 2021) target site C, respectively. We report the highest inference time we observed for each dataset and model.

| Method | VA | RandAug | GPS* | **OptTTA** | **OptTTA (M=2)** |
|---|---|---|---|---|---|
| Spinal Cord $(26 \leq D \leq 28)$ | 37.83 | 39.35 | 178.21 | 122.12 | 64.60 |
| Prostate $(D = 24)$ | 29.99 | 30.03 | 158.60 | 110.12 | 61.90 |
| Heart $(5 \leq D \leq 13)$ | 17.31 | 17.21 | 92.85 | 82.12 | 53.68 |

single 2D slice image takes 0.007 seconds. Thus, the time complexity to process $N$ test image volumes is $1 * T_{explore} + (N - 1) * T_{exploit}$, where $T_{exploit} << T_{explore}$. In practice, we use $|\mathcal{S}| = 21$ different sub-policies in the Exploration phase, $M = 128$ augmented images, and $k$ (=3) optimal sub-policies for inference, respectively. For example, on the Prostate dataset ($D = 24$), the search phase takes approximately 60 minutes followed by 110 seconds per subsequent image volume during the Exploitation phase, which is relatively fast. Furthermore, we can decrease OptTTA computational time to 61.90 seconds by setting M=2, while achieving similar performance (*cf.* Figure 7, Appendix D.2). As shown in Table 2, OptTTA is faster than the policy search method, namely GPS, in terms of inference time. On the other side, VA and RandAug are about 2x to 4x faster as unlike OptTTA, these methods do not involve learning optimal sub-policies. Nonetheless, they perform poorly under large domain shifts in terms of Dice score and Hausdorff distance (*cf.* Table 1, Section 3.2 and Table 8, Appendix D.5). For these reasons, we believe that OptTTA offers an excellent computational time/accuracy trade-off compared to the TTA baselines.

## Appendix B. Description of the Datasets and Pre-Processing

This section provides additional details about the three MRI datasets along with the pre-processing steps used in this paper.

### B.1. Spinal Cord Grey Matter Segmentation (SCGM) (Prados et al., 2017)

This is a multi-center and multi-vendor dataset of spinal cord anatomical images of healthy subjects from four different centers or sites (1, 2, 3, 4) and four MRI vendors (3 T Philips Achieva MRI system, 3 T Siemens TIM Trio, 3 T Siemens Skyra MRI scanner, 3 T whole-body Philips scanner) respectively. Each site contains images from 20 healthy subjects, out of which ten subjects have manual segmentation masks annotated by four experts. We use

label voting to merge these segmentation masks. The range of voxel resolutions varies from 0.25×0.25×2.5 mm to 0.5×0.5×5 mm, and the number of slices per volume ranges from 3 to 20. All the volumes were center cropped in the transverse plane with the crop size of 50mm and then resized to shape $256 \times 256$ pixels. The 2D slices in the transverse plane were used for training the segmentation model and inference. We use images from site 1 as the source domain while sites 2, 3, 4 are used as the target domain.

### B.2. Heart Image Segmentation Dataset (M&Ms) (Campello et al., 2021)

This dataset is composed of 375 patients with hypertrophic, dilated cardiomyopathies, and healthy subjects collected by six clinical centers from Spain, Canada, and Germany. As the data from the Canadian clinical center (# 6) is not publicly available, we use 340 patients in this work. The MRI scans come from four different vendors – A (Siemens) for center # 1, B (Philips) for center # 2 and 3, C (GE) for center # 4, and D (Canon) for center # 5. Each patient data is composed of several timestamped 3D volumes, out of which only a few timestamps (mostly 2) are annotated. In total, we use 190 annotated volumes from vendor A, 250 annotated volumes from vendor B, and 100 annotated volumes from vendor C and D. The range of voxel resolutions varies from 0.85×0.85×10 mm to 1.45×1.45×9.9 mm. All the volumes are first centered cropped to include only the heart region, followed by resizing the slices in the sagittal plane to $256 \times 256$ pixels. We use sagittal slices for training the segmentation model and inference. We use volumes from vendor A as the source domain and volumes from vendor B, C, D as the target domain.

### B.3. Prostate MRI Segmentation Dataset (Liu et al., 2020)

This is a multi-site dataset containing T2-weighted MRI for prostate anatomy with a segmentation mask collected from six different data sources out of three public datasets. The samples of site A, B are from NCI-ISBI 2013 dataset (Bloch et al., 2015), samples of site C are from Initiative for Collaborative Computer Vision Benchmarking (I2CVB) dataset (Lemaître et al., 2015), and sites D, E, F are from Prostate MR Image Segmentation 2012 (PROMISE12) dataset (Litjens et al., 2014). Following (Liu et al., 2020), we discard site C samples as they are mostly from unhealthy patients. Sites A, B, D, E, F contains 30, 30, 13, 12, 12 image volumes respectively. The volumes in this dataset are already centered cropped along the transverse plane with a size of $384 \times 384$ pixels used for training the segmentation model and inference. Sites A, B are used as the source domain, while sites D, E, F are used as the target domain.

## Appendix C. Uncertainty Metrics

The segmentation uncertainty can be evaluated by associating the model output's confidence with the correctness of the model predictions at the pixel level. Generally, strictly proper scoring rules are used to assess the calibration quality of predictive models (Gneiting and Raftery, 2007). We use three such metrics – Expected Calibration Error, Brier score and NLL (Gomariz et al., 2021). Table 3 provides the uncertainty measures of several methods computed on the Prostate dataset.

Table 3: Uncertainty analysis on the Prostate dataset.

| Method | ECE | Br | NLL |
|---|---|---|---|
| Source Model | 17.37 | 0.265 | 1.188 |
| TENT | 8.27 | 0.172 | 0.675 |
| RandAug | 9.60 | 0.204 | 0.723 |
| OptTTA | **3.49** | **0.139** | **0.528** |

**Expected Calibration Error (ECE) (lower is better).** The Expected Calibration Error analyzes the confidence values of test images predicted by the model versus their measured expected accuracy values. It measures whether the model is overconfident (high confidence and low accuracy) or under-confident (low confidence and high accuracy). For calculating the expected accuracy measurement, the pixels are put into $M$ bins according to their confidences predicted by the model, and the accuracy for each bin is computed. ECE is then calculated by summing up the weighted average of the differences between accuracy and the average confidence over the bins as follows:

$$ECE = \sum_{m=1}^{M} \frac{N_m}{N} \cdot |Acc(m) - Conf(m)| \tag{10}$$

where $N_m$ is the number of pixels, $Acc(m)$ is the average accuracy of pixels, $Conf(m)$ is the confidence of the $m^{\text{th}}$ bin, and $N$ is the total number of pixels.

**Brier score (Br) (lower is better).** The Brier score is a strictly proper score function that measures the accuracy of probabilistic predictions. It is equivalent to the mean squared error of the predicted probabilities with respect to ground truth. For a collection of $C$ possible segmentation classes, and $N$ pixels, Br metric can be computed as:

$$Br = \frac{1}{N} \sum_{i=1}^{N} \frac{1}{C} \sum_{c=1}^{C} [p(\hat{y}_i = y_c | x_i) - (\hat{y}_i = y_c)]^2 \tag{11}$$

**NLL (lower is better).** This metric measures the joint probability of observed data and can be used to estimate the uncertainty of the model predictions.

$$NLL = -\frac{1}{N} \sum_{i=1}^{N} \sum_{c=1}^{C} \ln(p(\hat{y}_i = y_c | x_i)) \cdot (\hat{y}_i = y_c) \tag{12}$$

where $p(\hat{y}_i = y_c | x_i)$ is the output confidence of the model for the class $y_c$ and input $x_i$.

### C.1. Aleatoric Uncertainty

We evaluate OptTTA in terms of aleatoric uncertainty estimation (Wang et al., 2019). This experiment shows that learning an optimal TTA policy by OptTTA further refines aleatoric uncertainty estimation than other TTA baselines like VA and RandAug. In particular, the dashed ellipses in Fig. 5 show that OptTTA leads to a lower error rate (occurrence) of overconfident incorrect predictions than other TTA baselines. Moreover, our joint histogram

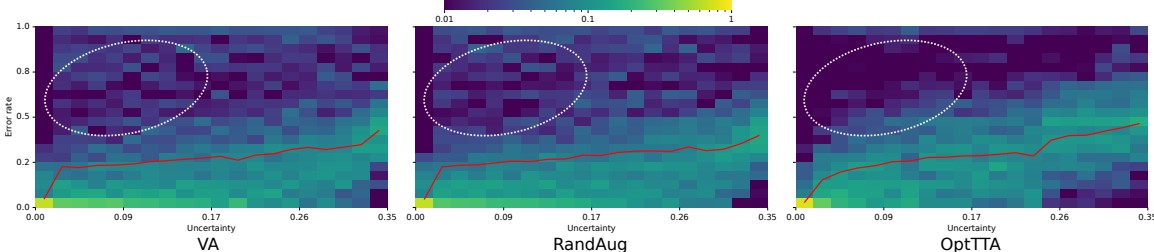

Figure 5: Normalized joint histogram of uncertainty estimation and error rate on the Prostate dataset. Given a pixel-wise uncertainty level (x-axis), we associate the frequency of pixel error rates along with the slices (y-axis). The red curve represents the mean error rate per uncertainty bin and dashed ellipses highlight the frequency of high error rates on different levels of overconfident predictions from VA, RandAug, and OptTTA.

is less noisy and shows an apparent monotonic increase of the error rate with respect to the uncertainty. These observations witness the efficiency of our learnable TTA policy framework in estimating the uncertainty under a domain shift scenario.

## Appendix D. Ablations

In this section, we present several ablation studies for the proposed method OptTTA on the Prostate dataset concerning sub-policy optimization criterion $\mathcal{L}_{\mathrm{OptTTA}}$, effects of exploration and exploitation, and test time performance based on the source model training strategy.

### D.1. Ablation Study on the Hyper-parameters of Loss Terms

We provide a sensitivity test of the hyper-parameters of the optimization criterion of OptTTA on the segmentation accuracy. Table 4 shows the effect of changing hyper-parameters of individual loss terms of $\mathcal{L}_{\mathrm{OptTTA}}$ by order of magnitude 10. We observe that the final segmentation accuracy is slightly sensitive to $\alpha_2$. It demonstrates that Entropy of Class Marginal ($\mathcal{L}_{\mathrm{cm}}$) is an important loss term for improving accuracy without supervision. However, it forces uniform prediction and degrades performance as its value increases. On the other hand, the segmentation accuracy is not very sensitive to $\alpha_1$ on average. We observe that penalizing the BN statistics discrepancy helps the Spinal Cord and Heart Datasets but slightly harms the Prostate dataset. This implies that our method can be applied to model architectures without BN layers.

### D.2. Exploration vs. Exploitation

In this subsection, we conduct several ablation experiments on the Exploration and Exploitation phases of OptTTA to justify the choice of sub-policy selection criterion, size of a sub-policy, and number of target domain images necessary for exploration.

Table 4: Sensitivity test with respect to hyper-parameters of $\mathcal{L}_{\text{OptTTA}}$.

| Hyperparameters | | Dice (%) | | | |
|---|---|---|---|---|---|
| $\alpha_1$ | $\alpha_2$ | Spinal Cord | Prostate | Heart | Average |
| 0.01 | 0.005 | $85.2_{\pm 3.6}$ | $83.0_{\pm 7.5}$ | $\mathbf{88.4}_{\pm \mathbf{3.6}}$ | $\mathbf{85.5}$ |
| 0 | | $84.1_{\pm 4.8}$ | $\mathbf{83.5}_{\pm 7.2}$ | $87.9_{\pm 4.5}$ | 85.2 |
| 0.001 | 0.005 | $84.3_{\pm 3.9}$ | $83.5_{\pm 8.6}$ | $88.0_{\pm 4.5}$ | 85.3 |
| 0.1 | | $84.8_{\pm 4.1}$ | $81.0_{\pm 9.6}$ | $87.9_{\pm 4.5}$ | 84.6 |
| | 0 | $\mathbf{85.3}_{\pm \mathbf{3.8}}$ | $80.5_{\pm 11.4}$ | $88.0_{\pm 3.4}$ | 84.6 |
| 0.01 | 0.0005 | $\mathbf{85.3}_{\pm \mathbf{3.8}}$ | $80.9_{\pm 10.5}$ | $88.3_{\pm 3.4}$ | 84.8 |
| | 0.05 | $81.7_{\pm 5.3}$ | $78.1_{\pm 9.6}$ | $86.4_{\pm 6.1}$ | 82.1 |

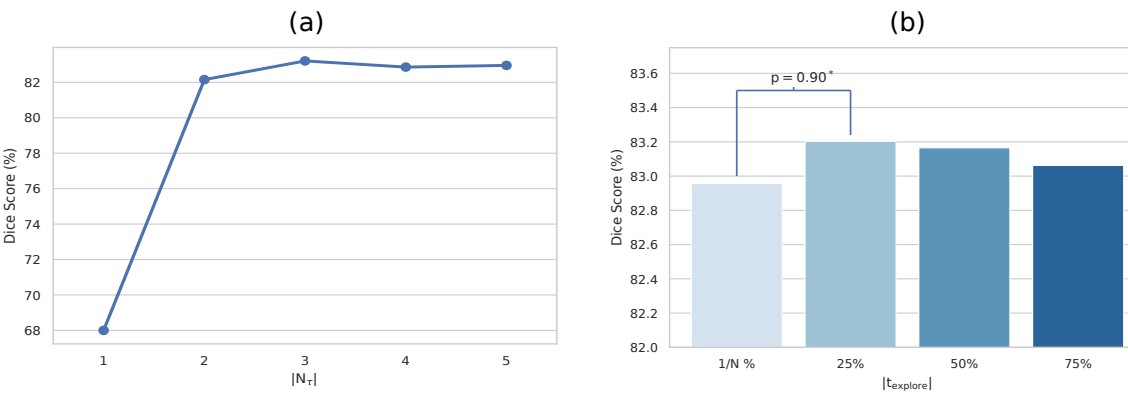

[*]Paired t-test with respect to the top result.

Figure 6: (a) Dice (%) vs. $|\mathcal{N}_\tau|$ on the Prostate dataset. (b) Dice (%) vs. $|t_{\text{explore}}|$ on the Prostate dataset. N is the size of the target site.

**Ablation on Selecting Top-k Sub-Policies.**    The last and crucial step of the Exploration phase is selecting the sub-set $\mathcal{T}^*$ comprising the top-k sub-policies from the set of optimized sub-policies $\mathcal{S}^*$. Table 5 shows the segmentation accuracy when different loss terms are used as the selection metric for the top-k sub-policies. We observe that $\mathcal{L}_{\text{OptTTA}}$ is the best choice for selection.

**Ablation on the Number of Augmentations Used in a Sub-Policy.**    Fig. 6 (a) shows the effect of changing the maximum size $N_\tau$ of sub-policy $\tau$. We observe that concatenating various augmentation operations helps in generating source-like augmented images and higher segmentation accuracy.

**Ablation on the Number of top-k Sub-Policies in the Exploitation Phase.**    Table 6 shows the effect of changing the number of top-k policies on the Spinal Cord dataset for sites 1 to 3. We observe that including all sub-policies degrades performance.

Table 5: Effect of changing sub-policy selection metric on the Prostate dataset. Using $\mathcal{L}_{OptTTA}$ or $\mathcal{L}_{bn}$ leads to similar performance while using $\mathcal{L}_{ent}$ alone degrades performance.

| Selection Metric | $\mathcal{L}_{\text{OptTTA}}$ | $\mathcal{L}_{\text{bn}}$ | $\mathcal{L}_{\text{ent}}$ |
|:---:|:---:|:---:|:---:|
| Dice (%) | $83.0_{\pm7.5}$ | $82.7_{\pm7.6}$ | $63.3_{\pm25.6}$ |

Table 6: Ablation experiment on the number of Top-k sub-policies in the Exploitation phase on the Spinal Cord dataset, with the source site=1 and target site=3.

| k | 1 | 2 | 3 | 5 | 10 | 15 | 21 |
|:---:|:---:|:---:|:---:|:---:|:---:|:---:|:---:|
| Dice (%) | $81.1_{\pm3.4}$ | $81.3_{\pm3.2}$ | $82.0_{\pm2.7}$ | $80.9_{\pm4.6}$ | $81.4_{\pm4.0}$ | $81.6_{\pm3.4}$ | $80.1_{\pm5.4}$ |

**Ablation on the Number of Augmented Views ($M$).** As shown in Fig. 7, increasing the number of augmented views leads to higher prediction accuracy for VA and RandAug. We observe that we reach a plateau at $M = 32$. On the other hand, OptTTA seems less sensitive to values of $M$, having a similar performance by generating 128 or only two views. These observations support learning an optimal augmentation policy by TTA methods.

**Ablation Study on the Number of Images used for Exploration ($|t_{\textbf{explore}}|$).** In practice, we do not have access to all test time data at once. However, we can fine-tune the optimal sub-policies found in the exploration phase on the test images in an online manner. Since exploration is expensive to compute for every test image, we benefit by directly applying the optimal sub-policies found during the exploration phase, thus making inference faster. Fig. 6 (b) shows the effect of exploring more than one target image on the overall Dice score for the Prostate dataset for sites A, B to F.

### D.3. Performance Comparison of TTA Methods Under Different Training Strategies of the Source Model

The performance of TTA methods often relies on the initial source model performance, considering these methods' limitations. Fig. 8 shows correlational analysis supporting that the accuracy of TTA methods depends on the augmentation policy used for training as well as the training dataset size. Nevertheless, OptTTA still surpasses the baselines on these particular settings showing our method is more robust under these training setup variations.

### D.4. Performance Comparison of Baselines Using Multiple Source Domains

We also show the quantitative comparison results (Dice (%)) with state-of-the-art methods on the Spinal Cord dataset and trained on multiple source domains. Overall, aggregating information from diverse source domains can improve the model's generalization capability compared to the models trained on a single source only. As shown in Table 7, OptTTA

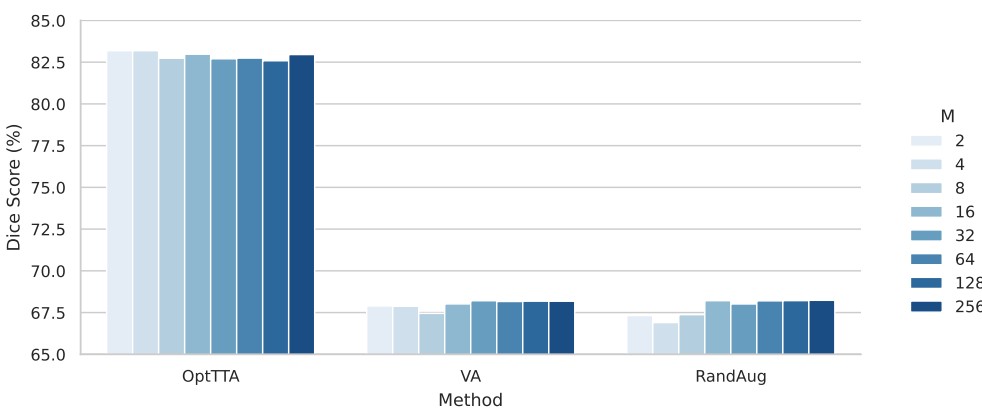

Figure 7: Dice (%) scores vs. $M$ values on the Prostate dataset for various TTA methods, including OptTTA, VA, and RandAug.

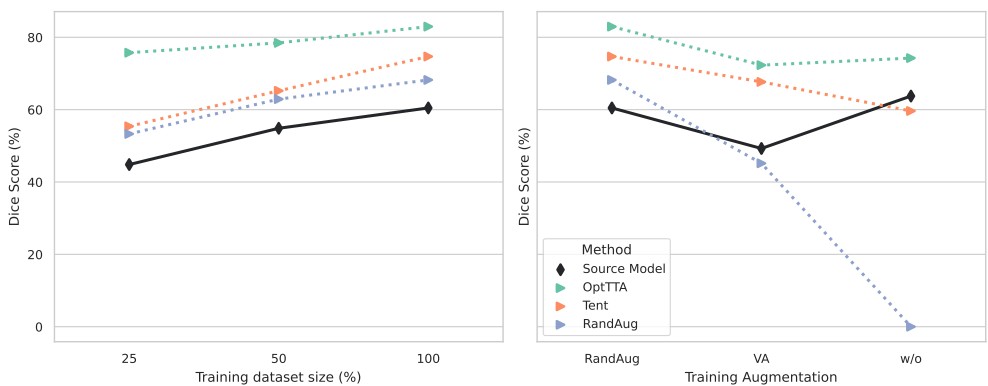

Figure 8: Correlation analysis of the Dice (%) scores vs. training data size and training augmentation policy on the Prostate dataset.

Table 7: Dice (%) results of mean(±std) on the Spinal Cord dataset. The models are trained on multiple source domains. The largest domain gap w.r.t. source domain is highlighted in red, and Bold values denote the best performances.

| Source site(s) | Target site(s) | Lower bound | UDA | | TTMA | | | TTA | | | |
|---|---|---|---|---|---|---|---|---|---|---|---|
| | | DeepAll | ADVENT | ProDA | BN | TENT | PL | VA | RandAug | GPS* | **OptTTA** |
| 2,3,4 | 1 | **88.0**±2.7 | 86.0±4.3 | 87.7±2.9 | 86.3±3.2 | 86.7±3.2 | 86.4±3.1 | 86.5±3.0 | 85.8±3.3 | 85.6±3.2 | 87.3±2.2 |
| 1,3,4 | 2 | **88.3**±0.7 | 87.6±0.9 | 87.9±0.8 | 87.2±0.9 | 87.2±0.7 | 87.1±0.8 | 87.3±0.9 | 87.3±0.9 | 87.8±0.8 | 88.1±0.6 |
| 1,2,4 | 3 | 50.5±28.3 | 85.8±1.8 | 78.2±2.5 | 69.5±5.6 | 74.4±2.3 | 71.9±5.0 | 48.3±28.5 | 45.6±27.2 | 70.0±16.7 | **87.0**±**2.0**‡ |
| 1,2,3 | 4 | **90.9**±1.1 | 88.5±2.3 | 90.8±1.0 | 90.5±1.2 | 90.3±1.3 | 90.4±1.2 | 90.0±0.9 | 89.7±0.9 | 89.8±1.2 | 90.0±0.8 |
| Average | | 79.4±21.9 | 87.0±2.9 | 86.1±5.2 | 83.4±8.8 | 84.7±6.4 | 84.0±7.7 | 78.0±22.4 | 77.1 ±22.8 | 83.3±12.5 | **88.1**±**2.0**‡ |

[‡] $p < 0.005$, [†] $0.005 < p < 0.05$: A paired t-test with respect to the top results.

significantly outperforms other TTA methods on average and shows marginal gains over state-of-the-art UDA methods while not using information about source domains. Additionally, we provide a new baseline, DeepAll, by aggregating all source domains data followed by segmentation model standard training. OptTTA achieves competitive performance on par with DeepAll in most cases without using knowledge from source domains. Nevertheless, in the presence of substantial domain shift (target site=3), OptTTA significantly improves the accuracy upon DeepALL, demonstrating our model's generalization aspects under large domain shift.

### D.5. Additional Results

#### D.5.1. Hausdorff Distance Metric and Qualitative Results

Table 8 compares the Harmonic Mean 95[th] percentile Hausdorff Distance (HD95) of the segmentation predicted by OptTTA against several baselines, while Fig. 9 and Fig. 10 show additional qualitative segmentation results on 2D slices and 3D volumes, respectively.

#### D.5.2. Evolution of Sub-Policies in Exploration and Exploitation Phases

**Exploration Phase.** Fig. 11 shows the evolution of 21 different sub-policies on a sample test image from the Spinal Cord dataset by a fixed step size of 80 iterations. We observe that the segmentation prediction of the source model on the target image improves as the parameters of a sub-policy are optimized using the loss $\mathcal{L}_{\text{OptTTA}}$ defined in Sec. 2.2.1. However, not all the sub-policies perform equally well at the end of optimization.

**Exploitation Phase.** Fig. 12 shows segmentation results on augmented views of the target image obtained by fine-tuning top-3 sub-policies $\mathcal{T}^*$ found in the Exploration phase.

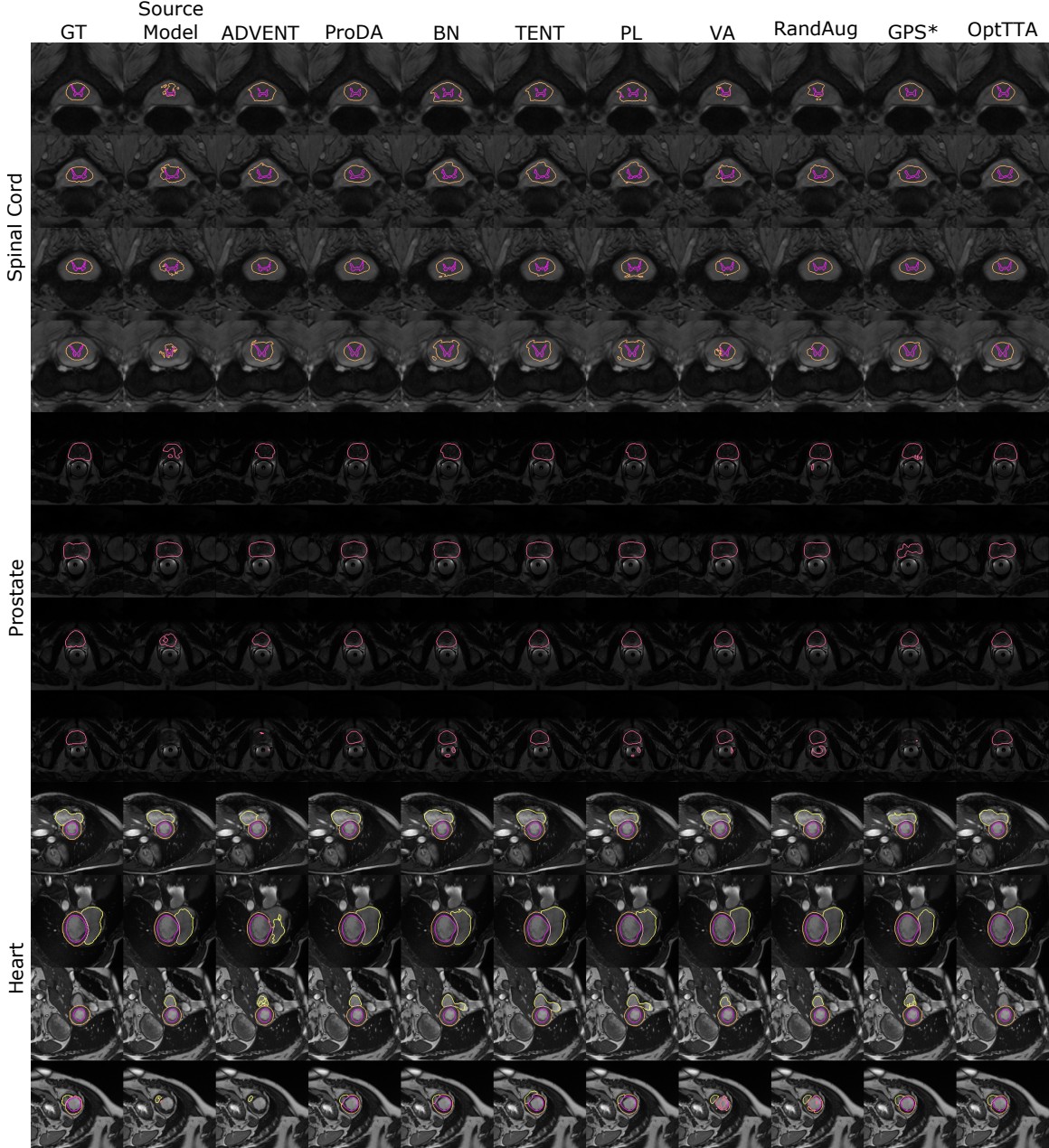

Figure 9: Qualitative segmentation results (2D slices) on three multi-center, multi-vendor MRI datasets: Spinal Cord (Prados et al., 2017), Prostate (Liu et al., 2020) and Heart dataset (Campello et al., 2021).

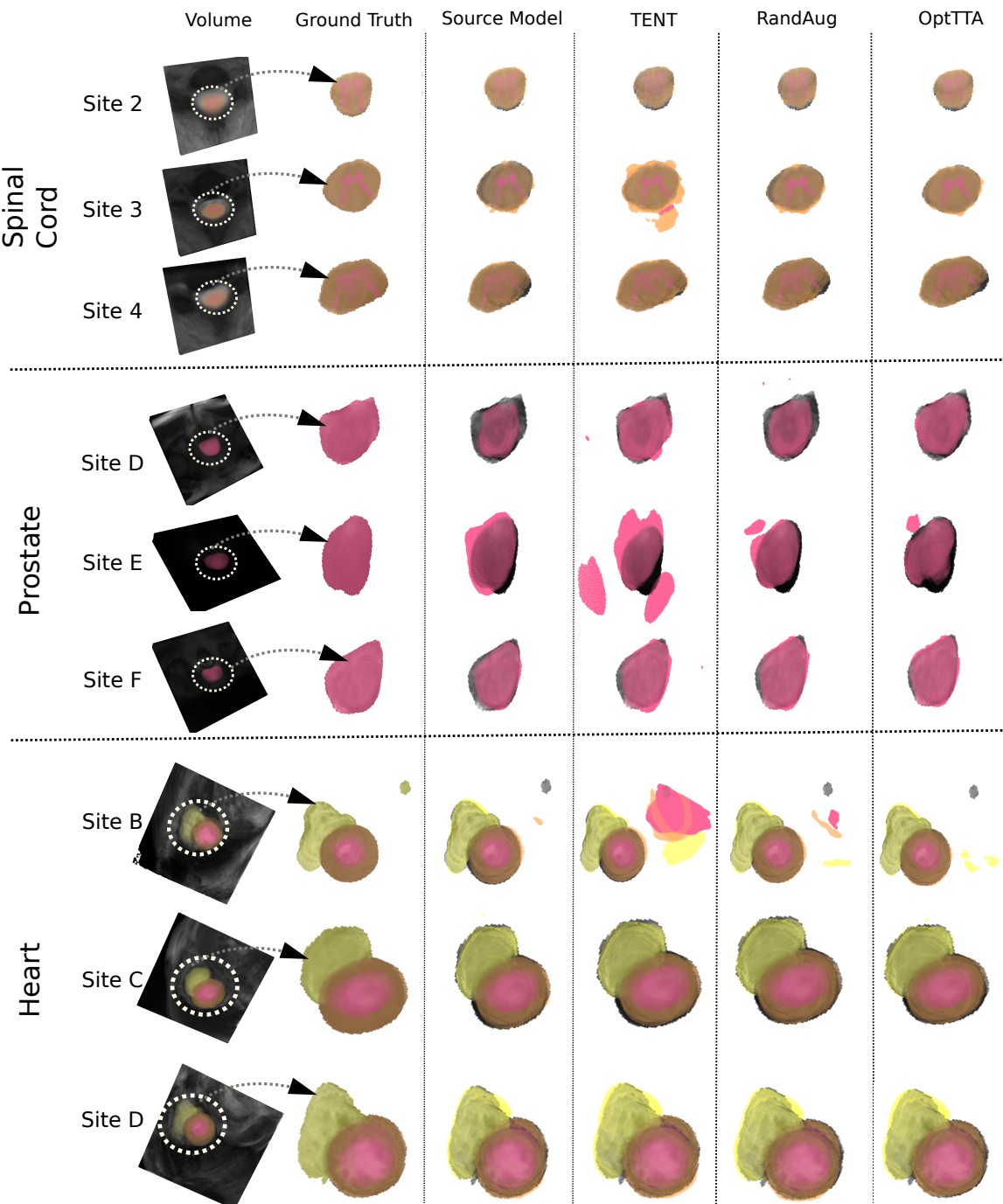

Figure 10: Qualitative 3D segmentation results on three multi-center, multi-vendor MRI datasets: Spinal Cord (Prados et al., 2017), Prostate (Liu et al., 2020) and Heart dataset (Campello et al., 2021).

Number of iterations

Sub-policies

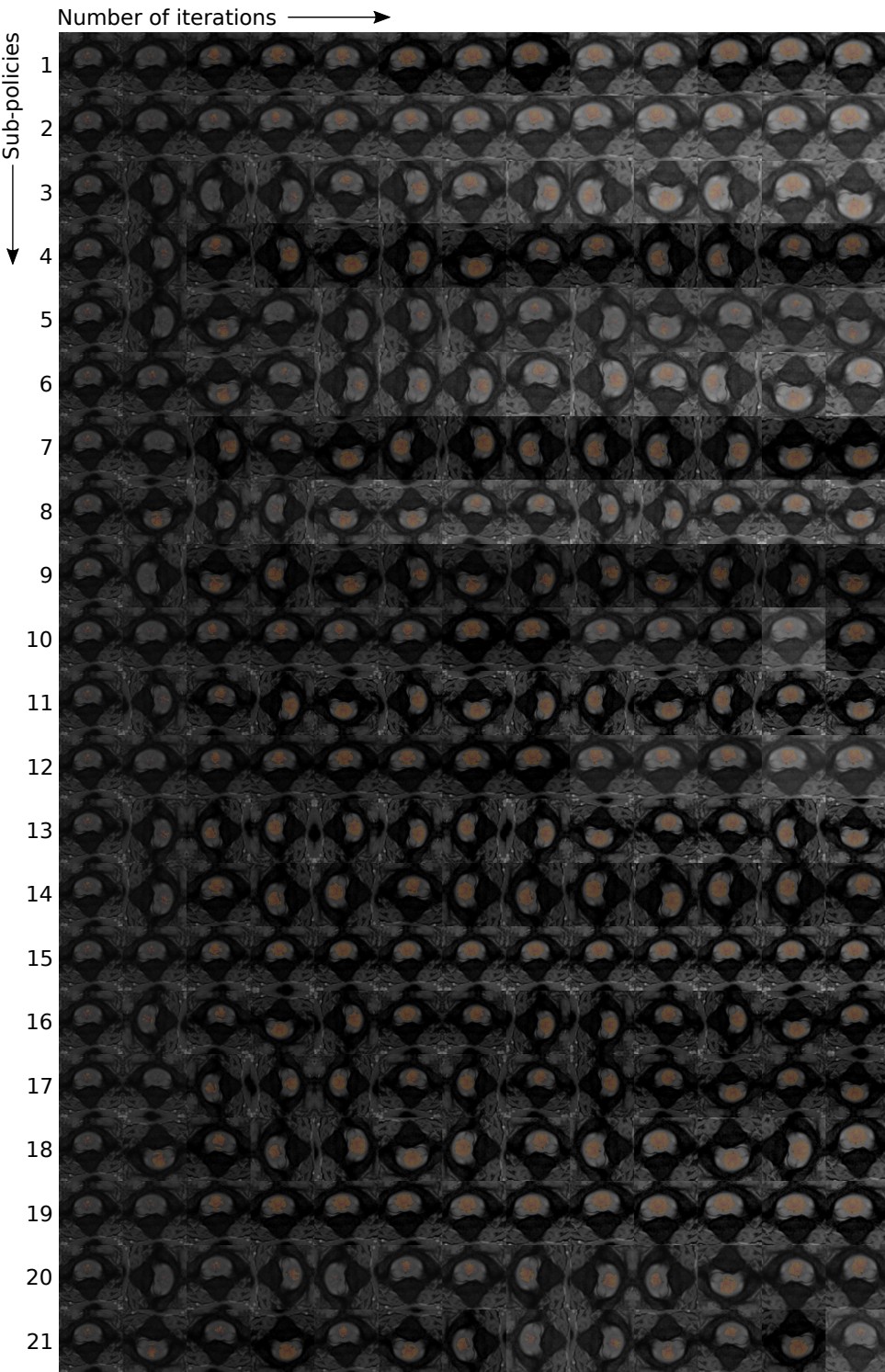

Figure 11: Evolution of 21 sub-policies during the Exploration phase on the Spinal Cord dataset.

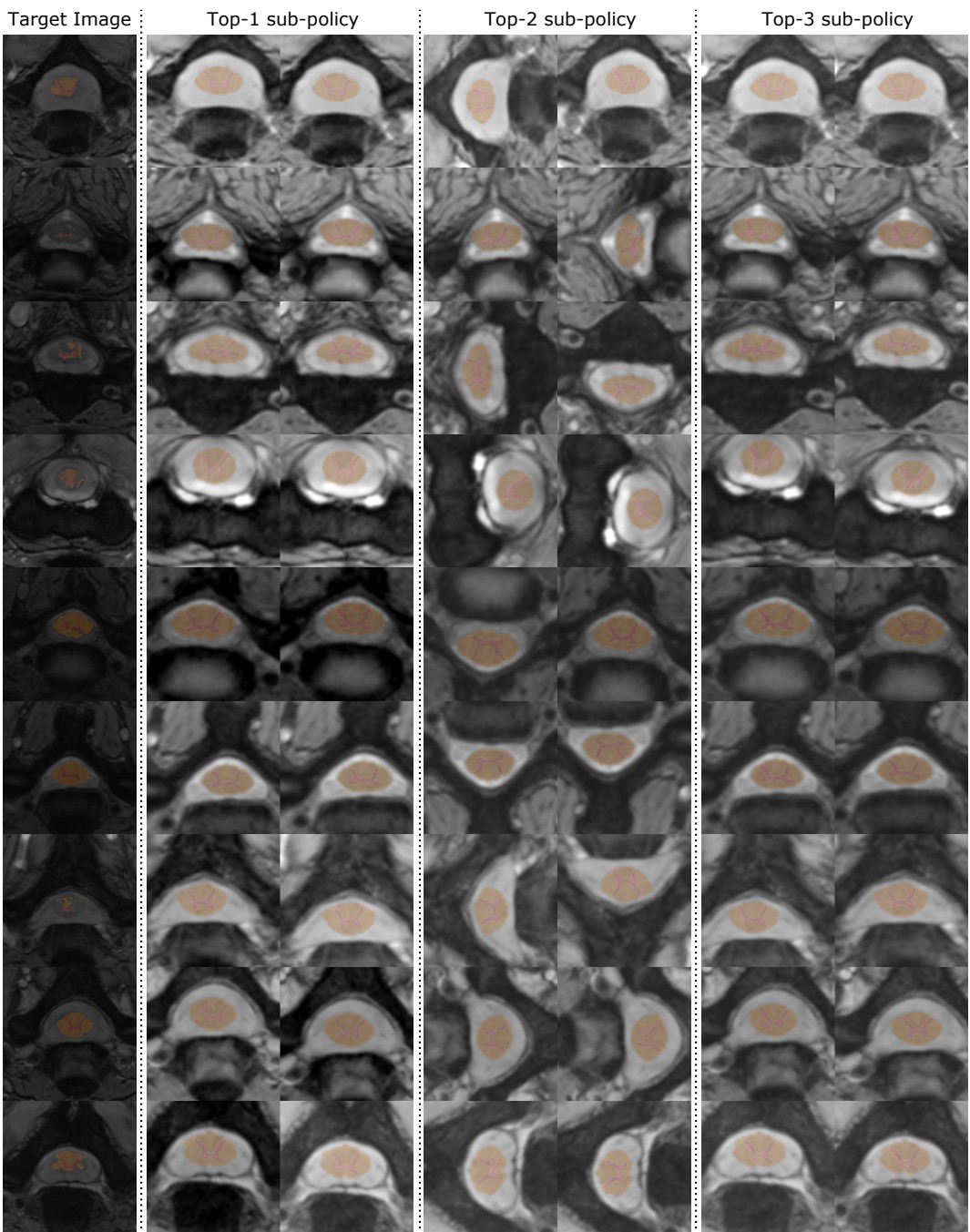

Figure 12: Exploiting top-3 sub-policies after the Exploration phase on the Spinal Cord dataset. The first column shows the segmentation prediction of the source model directly on the target image. The next three columns show the augmented views of the target image generated by the top-1, top-2 and top-3 policies in the Exploitation phase and their corresponding segmentations predicted by the source model.

Table 8: Harmonic Mean 95$^{\text{th}}$ percentile Hausdorff Distance (HD95) in pixel (px) on datasets: Spinal Cord (Prados et al., 2017), Prostate (Liu et al., 2020), and Heart dataset (Campello et al., 2021). The largest domain gap w.r.t. source domain is highlighted in red, and bold values denote the best performances.

| | | Lower Bound | UDA | | TTMA | | | TTA | | | |
|---|---|---|---|---|---|---|---|---|---|---|---|
| Target site | # Volumes | Source Model | ADVENT | ProDA | BN | TENT | PL | VA | RandAug | GPS* | OptTTA |
| (Source Site 1) | | | | | | Spinal Cord | | | | | |
| 2 | 10 | 1.71 | 1.34 | 1.19 | **1.15** | 1.16 | **1.15** | 1.63 | 1.36 | 1.39 | 1.16 |
| 3 | 10 | 3.05 | 2.46 | 2.32 | 68.48 | 67.33 | 67.87 | 2.90 | 2.86 | 2.37 | **2.00** |
| 4 | 10 | 1.26 | 1.17 | **1.04** | 1.07 | 1.07 | 1.07 | 1.26 | 1.16 | 1.20 | 1.05 |
| Harmonic Average | | 1.76 | 1.50 | 1.34 | 1.65 | 1.65 | 1.65 | 1.71 | 1.54 | 1.52 | **1.30** |
| (Source Sites A,B) | | | | | | Prostate | | | | | |
| D | 13 | 3.59 | 3.74 | 2.93 | 9.91 | 6.83 | 9.44 | 3.47 | 3.63 | 3.66 | **2.18** |
| E | 12 | 10.31 | 7.70 | **3.56** | 18.31 | 7.50 | 17.02 | 6.17 | 7.62 | 6.43 | 4.18 |
| F | 12 | 7.37 | 6.59 | 3.68 | 8.18 | 4.13 | 7.09 | 4.99 | 5.75 | 6.12 | **2.69** |
| Harmonic Average | | 5.77 | 5.40 | 3.34 | 10.77 | 5.77 | 9.80 | 4.57 | 5.11 | 5.01 | **2.79** |
| (Source Site A) | | | | | | Heart | | | | | |
| B | 250 | 1.30 | 1.29 | 1.26 | 1.82 | 3.08 | 1.80 | 1.27 | 1.27 | 1.71 | **1.23** |
| C | 100 | 1.54 | 1.65 | 1.48 | 2.39 | 4.25 | 2.33 | 1.38 | 1.39 | 1.82 | **1.34** |
| D | 100 | 1.48 | 1.57 | 1.41 | 2.58 | 6.14 | 2.55 | **1.26** | **1.26** | 1.67 | 1.27 |
| Harmonic Average | | 1.39 | 1.41 | 1.34 | 2.06 | 3.72 | 2.04 | 1.29 | 1.29 | 1.73 | **1.26** |

