# OpenReview forum: "OptTTA: Learnable Test-Time Augmentation for Source-Free Medical Image Segmentation Under Domain Shift"
_MIDL.io/2022/Conference — MIDL 2022_

### Official Review · Reviewer_Bc31 · 2022-01-24

**Confidence:** 5
**Preliminary Rating:** 3
**Recommendation:** Poster

**Summary:**

The paper proposes a method for domain adaptation that does not require access to source data. This is important because source domain data is often not available at the target site location due to data protection issues. It achieves this by learning target-data augmentation policies that match the source data entropy and batch norm stats.

**Strengths:**

. Interesting method for DA that does not require the source domain data, thus making it more practical and feasible.
. The proposed policy search approach is interesting, novel and appears to be robust
. The statistical analysis of the results is quite strong and proper
. The results on the TTA uncertainty estimation are convincing and show significant improvements

**Weaknesses:**

. The search space over policies is quite restricted, meaning that the proposed method can only domain adapt to relatively small gaps.
. The divergence loss seems non-optimal from a label statistics point of view.
. The choice of datasets is not good for a domain adaptation experiment, as the domain gap is small for all experiments
. The baseline network is 2D and very simple.
. No details are provided on the source training augmentation regime.

**Deanonymize Review:**

no

**Detailed Comments:**

Below are comments provided in reading order:
. Section 1 - "we propose the first learnable TTA" there are several previous methods for test-time augmentation, including one-shot methods, applied to medical imaging data. The authors might argue that the word "learnable" make the statement true, but the word learnable here is to imprecise.

. Section 2.1 -  The Policy search space comprises mostly geometrical transformations, a few very simple intensity transformation, and gaussian blur. These are not sufficient for many domain adaptation problems and only work on approximately the same modality. For example, previous MICCAI challenges on DA have tried T1->T2 and T1->CT domain gaps, which would not be covered by these policies.

. Section 2.2. - The divergence loss encourages predictions to be uniformly distributed, which is relies on a wrong assumption. Why would classes be equally distributed? If a class is very unlikely/small, the DIV loss would attempt to grow this region. Why not match the probability distribution of the classes between source and target domains?

. Section 3.1 - The choice of datasets is good from a public availablability point of view, but non-ideal from a domain adaptation point of view. The different domains for all datasets are only different sites or scanner manufacturers, and not different sequences/modalities, meaning that the domain gap is quite small.

. Section 3.1 - The choice of a 2D Unet rather than a 3D unet is again non-ideal, mainly given that the "large variance among data from different centres" reason that is provided to justify the 2D architecture is also the aspect (domain gap) that is being tackled in this work.

. It is not clear how strong is the augmentation on the source domain for all baseline models. Given that the domain gap is quite small, strong augmentation of the source domain can make models that are robust to small domain shifts. Would a well augmented model (eg. a strong nnUNET baseline) trained on the source data perform much worse on the target domains? Authors should try and train such a model on the data in order to get a strong baseline that is known to be well validated.

. How was the data partitioned for the ablation study? Was the ablation study done on the same partitions as the main experiments? And if so, how can the authors guarantee that no bias in introduced?

**Final Rating After The Rebuttal:**

4: Weak Accept

**Justification Of The Final Rating:**

The authors have addressed some of my concerns, but I remain unconvinced about the degree of domain adaptation and its utility. If the domain gap is small (same modality but different scanners) as proposed here, then augmentation techniques, destination methods, and many other approaches can be seen as domain adaptation methodologies. How does the proposed method perform against models with robust augmentation stacks (e.g. NNUNET, etc)? Given that the proposed method was only validated in 2D, how does it compare to simpler models that work well in 3D? It is complex to assess the value of the proposed contributions given that the comparison baselines have not been chosen to demonstrate the value of the proposed improvements against strong SOTA models that are expected to perform well under the data the model has been exposed to.

**Paper Type:**

methodological development

**Questions To Address In The Rebuttal:**

. How would the model work on larger domain gaps?
. How would a strong baseline model work (e.g. nnUNET) in comparison with the proposed method?
. How much does the L_div actually contribute to performance? A alpha of 0.005 is very small.

**Special Issue:**

no

---

### Official Review · Reviewer_istg · 2022-01-24

**Confidence:** 3
**Preliminary Rating:** 4
**Recommendation:** Oral

**Summary:**

This paper propooses a technique to learn a Test Time Augmentation policy that leads to improved segmentation performance under domain shifts. The idea is to look in the search space of possible augmentation operations for policies that results in augmented image views whose segmentations are then evaluated *without the need of ground-truth segmentations*. How, one may ask? By making sure that batch-norm statistics of a batch of augmented images are similar to batch-norm statistics of the model trained on source data, and enforcing confident predictions (typical of source data), plus pushing the solutions to contains pixels of many segmentation classes (to avoid degenerate solutions). Since all these three items do not require access to the training data itself (only batch-norm statistics of the trained model), this allows to evaluate the performance of TTA sub-policies, thereby enabling their optimization in test time. Once the optimial sub-policy is found (and this is done in an per-image basis), one then generates several segmentations and ensembles them together, resulting in enhanced performance, as shown in the experimental section.

**Strengths:**

I find this paper to be really well-polished and strong. The main strength in my opinion is the experimental analysis, comprising many different domain shift scenarios, and analyzing several aspects of the problem. It is only mentioned very briefly in the introduction, but it is also a nice features that the method does not need access to training data and can therefore respect data privacy after being trained.

**Weaknesses:**

For all the detail in the paper (24 pages, with 12 figures, 6 tables, 4 appendices, and 9 sub-appendices, I find really surprising that there is no mention to the computational price that the user has to pay in order to process a single MRI scan with this approach. TTA techniques are usually criticized because they incur in an extra inference cost of generating N predictions when before you just generated one. Now we are doing this plus solving a policy search problem *per slice*. If the user is going to spend more time finding an optimal set of augmentations than training the initial segmentation model, it sounds like a huge price to pay for what in some cases is a handful of dice score points (like in the heart experiment). I would really like the authors to provide an analysis on the average computational time that their method requires per scan, as well as a comparison with other approaches from the point of view of complexity and resource consumption to generate the improved segmentations.

**Deanonymize Review:**

yes

**Detailed Comments:**

* "we observed those self-training techniques often produce incorrect predictions in the presence of large domain shifts leading to error accumulation during test-time model adaptation". Is this claim is supported by any experiment? I would advice to remove it, as it sounds a bit speculative (like an opinion).
* More in general, in the related work the authors go over previous papers on TTA like in a laundry list, criticizing each of them in what I find like a subjective point of view, like for example "Notably, the arbitrarily chosen augmentations may be too compact to cope with a large domain gap." And in particular, a critic of the authors to other learned loss for TTA previous works is that they require a validation set to be applied. However, I guess they have also looked into some kind of validation data in order to decide on the alpha=0.01, alpha2=0.005 magic values, plus all the other hyper-parameters described in page 6. I would advice to either lower the critical tone on previous works, or demonstrate with experiments the shortcomings that are being pointed out in this section.
* in building the policy search space of section 2.1, do we not need some basic constraints? Like, do not apply horizontal flip followed by horizontal flip, or defining a maximum amount of operations?
* I am curious about if we need to build a number of augmented views that is consistent with (i.e. the same as) the batch-size used during training, so that matched batch-norm statistics like in eq. (3) are meaningful? Or is it ok to find a sub-policy that tells you to only use, say, two augmented images, even if batch-size was 8 during training?
Questions To Address In The Rebuttal:
* Regarding calibration evaluation, I find it a bit too qualitative. Like, only reliability diagrams and box-plots are shown, plus the ellipses of Figure 5, but no quantitative (numbers) analysis is given. It might be easier for the reader to have a table with numbers, I think. Usually one expects to find Expected Calibration Errors, or some derivation of this metric that is more suitable for medical image segmentation (see Orthogonal Ensemble Networks for Biomedical Image Segmentation, AJ Larrazabal, C Martínez, J Dolz, E Ferrante, MICCAI 2021). Or maybe there is a way to quantify those ellipses similar to the Wang 2019 paper? Like, add up the values of the 2d histogram over the diagonal?

**Final Rating After The Rebuttal:**

5: Strong Accept

**Justification Of The Final Rating:**

It seems to me that the authors addressed all the points I raised in my initial review. The computational complexity analysis provided to the other reviewer that asked for it is thorough and complete, I believe this paper should be accepted to MIDL.

**Paper Type:**

methodological development

**Questions To Address In The Rebuttal:**

Aside from the detailed comments, what I would like to see the most is an analysis on computational complexity and what is the overload in terms of computing time per generated segmentation that the user suffers from using the proposed approach, when compared to other methods.

**Special Issue:**

yes

---

### Official Review · Reviewer_qKzC · 2022-01-25

**Confidence:** 4
**Preliminary Rating:** 5
**Recommendation:** Best Paper Award, Oral, Poster

**Summary:**

The paper presents a learnable test-time augmentation method for tackling the problem of domain shift in medical image segmentation without access to source data. In an exploration mode, the method finds an augmentation policy combining different image transformations which reduces the network's prediction entropy and distance to source batch normalization statistics, on a small set of test images. In exploitation mode, this policy is fine-tuned for each image. The predictions obtained with the top K best policies are averaged to get the final segmentation. Comprehensive experiments on several datasets with different characteristics show the advantage of the method in terms of segmentation accuracy and uncertainty calibration, compared to recent unsupervised domain adaptation and test-time adaptation approaches.

**Strengths:**

* The proposed approach, using a differentiable strategy to find good augmentation policies, is quite original and could be very useful in practice.

* The experimental validation is quite extensive, and clearly demonstrates the advantage of the method compared to recent adaptation approaches, including ones that have access to source data.

* The paper is well written and clearly presents the related work, contributions, method, experiments, and results.


**Weaknesses:**

* This is not a weakness per se, but the paper is quite long (24 pages in total with Appendices) and a lot of important material, such as the details of the optimization algorithm and key ablation studies, is found in Appendices.

* The paper also could be improved by providing and discussing results on the computational complexity of the proposed method. I believe this is an important element since the method requires some optimization at the time.


**Deanonymize Review:**

no

**Detailed Comments:**

Other comments:

* In Eq (3), I wonder if a weight would be necessary to balance the terms corresponding to the mean and variance? Are the value ranges of the two terms comparable? Why not consider a more theoretically-grounded distance measure such as the KL divergence between two Gaussians.

* The loss term in Eq (5) could be better presented/motivated. The name Divergence loss does not really represent what is measured, which is the entropy of class marginals. Authors could also mention that the combination of (4) and (5) corresponds to mutual information, as explained in (Bateson et al., 2021).

* It would also be useful to mention which augmentation can be used in the method. For example, are there any transformations for which the parameters cannot be optimized through gradient descent?

* The proposed method boosts the segmentation accuracy by combining the prediction obtained with different augmentation policies. However, it is well-known that ensemble learning requires some diversity across the ensembled models/predictions. Can the authors evaluate/discuss the diversity of the different augmentations and how this diversity impacts the performance?

**Final Rating After The Rebuttal:**

5: Strong Accept

**Justification Of The Final Rating:**

My main concern was the lack of analysis on computational complexity. The authors' response for this point was very detailed and clear. Despite the added complexity of the proposed TTA method, I still believe it is a nice contribution to the medical imaging community. Hence, I keep my overall score of Strong Accept.

**Paper Type:**

methodological development

**Questions To Address In The Rebuttal:**

Evaluate/discuss the computational overhead of the method in test time. Explain the balancing of the two terms in the BN loss. Better motivate the Diversity (marginal entropy) loss and why the marginal distribution of source examples cannot be used as target (e.g., minimizing a KL between the class marginals of source and test examples). Discuss the diversity of augmentations and how it affects the ensemble prediction.

**Special Issue:**

yes

---

### Meta-Review · Area_Chair_1ax6 · 2022-02-19

**Recommendation:** Accept (Oral)
**Confidence:** 5

**Metareview:**

This work initially received very positive comments, with two reviewers accepting the paper (strong and weak accept) and one reviewer giving borderline. After the rebuttal, two reviewers increased their score to strong and weak accept. Despite these final scores, there has been an interesting discussion between the authors and reviewer Bc31, where the main weakness is related to the domain gap size in the experiments. Just to shed some light on this concern and give my personal opinion, I am aware that domain shifts between the same image modalities, across scanners, could be large. Nevertheless, I agree with reviewer Bc31 that a cross-modality experiment (e.g., T1 to T2) would have been appreciated. Note for example, that these experiments are also common in Test-Time Adaptation in medical image segmentation (e.g., Karani et al., MedIA'21). Nevertheless, this does not affect the quality of this work and its potential impact.

---

### Decision · Program_Chairs · 2022-02-28

Accept